# Translational Research for Orthopedic Bone Graft Development

**DOI:** 10.3390/ma14154130

**Published:** 2021-07-24

**Authors:** Maria J. C. Vilela, Bruno J. A. Colaço, José Ventura, Fernando J. M. Monteiro, Christiane L. Salgado

**Affiliations:** 1Instituto de Investigação e Inovação em Saúde (i3S), Universidade do Porto, 4200-135 Porto, Portugal; jcaspurrovilela@gmail.com (M.J.C.V.); fjmont@ineb.up.pt (F.J.M.M.); 2Instituto Nacional de Engenharia Biomédica (INEB), 4200-135 Porto, Portugal; 3Instituto de Ciências Biomédicas Abel Salazar (ICBAS), Universidade do Porto, 4050-313 Porto, Portugal; 4Department of Animal Science, CECAV—Animal and Veterinary Research Centre UTAD, University of Trás-os-Montes and Alto Douro, 5000-801 Vila Real, Portugal; bcolaco@utad.pt; 5Artur Salgado S.A., 4250-288 Porto, Portugal; j.ventura@artursalgado.pt; 6Faculdade de Engenharia, Universidade do Porto, 4200-465 Porto, Portugal

**Keywords:** collagen, nanohydroxyapatite, scaffold, bone regeneration, biomaterials

## Abstract

Designing biomaterials for bone-substitute applications is still a challenge regarding the natural complex structure of hard tissues. Aiming at bone regeneration applications, scaffolds based on natural collagen and synthetic nanohydroxyapatite were developed, and they showed adequate mechanical and biological properties. The objective of this work was to perform and evaluate a scaled-up production process of this porous biocomposite scaffold, which promotes bone regeneration and works as a barrier for both fibrosis and the proliferation of scar tissue. The material was produced using a prototype bioreactor at an industrial scale, instead of laboratory production at the bench, in order to produce an appropriate medical device for the orthopedic market. Prototypes were produced in porous membranes that were e-beam irradiated (the sterilization process) and then analysed by scanning electron microscopy (SEM), confocal laser scanning microscopy (CLSM), dynamic mechanical analysis (DMA), cytotoxicity tests with mice fibroblasts (L929), human osteoblast-like cells (MG63) and human MSC osteogenic differentiation (HBMSC) with alkaline phosphatase (ALP) activity and qPCR for osteogenic gene expression. The prototypes were also implanted into critical-size bone defects (rabbits’ tibia) for 5 and 15 weeks, and after that were analysed by microCT and histology. The tests performed for the physical characterization of the materials showed the ability of the scaffolds to absorb and retain water-based solvents, as well as adequate mechanical resistance and viscoelastic properties. The cryogels had a heteroporous morphology with microporosity and macroporosity, which are essential conditions for the interaction between the cells and materials, and which consequently promote bone regeneration. Regarding the biological studies, all of the studied cryogels were non-cytotoxic by direct or indirect contact with cells. In fact, the scaffolds promoted the proliferation of the human MSCs, as well as the expression of the osteoblastic phenotype (osteogenic differentiation). The in vivo results showed bone tissue ingrowth and the materials’ degradation, filling the critical bone defect after 15 weeks. Before and after irradiation, the studied scaffolds showed similar properties when compared to the results published in the literature. In conclusion, the material production process upscaling was optimized and the obtained prototypes showed reproducible properties relative to the bench development, and should be able to be commercialized. Therefore, it was a successful effort to harness knowledge from the basic sciences to produce a new biomedical device and enhance human health and wellbeing.

## 1. Introduction

The extracellular matrix (ECM) is very important for cells’ microenvironments and survival. It not only gives structural support to cells and tissues but also provides signaling cues that regulate cells’ behaviour in multicellular organisms, such as cell growth, differentiation, shape and viability [1]. It is composed of structural proteins (i.e., collagen), polysaccharides/glycosaminoglycans (GAGs), and adhesion proteins, such as integrins, which make the connection between the ECM and the cells. These constituents are constantly being synthesized, secreted and exchanged between the cells and the ECM. In bone, the extracellular matrix is a key part of the tissues’ structure and function. It is more rigid than the ECM that is present in other tissues because of tissue mineralization, which is the deposition of calcium phosphate crystals, corresponding to the inorganic phase of bone [2]. Bone loss due to congenital defects, trauma, accident or infections, or after tumour resection are major clinical problems that need to be addressed [3,4]. The use of bone grafts is the most common procedure to treat these clinical problems, though there is the possibility of infection and implant rejection, as well as donor site morbidity. In addition, they are associated with high costs [3,5]. In order to try to avoid these major disadvantages, alternative materials need to be found that are able to mimic the mechanical properties, structure and functions of the bone tissue [6]. To this end, 3D scaffolds are widely used in bone regeneration to promote tissue growth by mimicking the ECM of the native tissues [7,8,9]. Ideal scaffolds should be biocompatible, biodegradable and non-immunogenic, should promote cell and material surface interactions to allow adhesion, should provide the diffusion of nutrients and other molecules through their structures, and should have suitable mechanical properties, depending on their final application. Thus, an adequate biomaterial for bone regeneration applications needs to be osteoinductive, osteoconductive and osteogenic [10]. To this end, 3D scaffolds can have different shapes and sizes, different fabrication techniques and different composition. Nowadays, some of the most common scaffolds are electrospinning fibers, decellularized tissues, microspheres, ceramics, hydrogels and cryogels. This last form of scaffold, with a composition of collagen and nanohydroxyapatite similar to the bone, was the one chosen in our work to be applied as a medical device in bone regeneration. The objective of this work was to perform and evaluate a scaled-up production process of a porous scaffold based on collagen and nanohydroxyapatite, which promotes bone regeneration and works as a barrier for both fibrosis and the proliferation of scar tissue. The scaffold was characterized in terms of its chemical, physical and mechanical properties, and showed similar characteristics to the native bone ECM. We also observed the in vitro and in vivo cell/tissue behavior within the different samples, and they showed biocompatibility, high cellular viability, proliferation and osteogenic differentiation, which is an advantage for application as a medical device in bone regeneration.

## 2. Materials and Methods

### 2.1. Materials 

Type I collagen from bovine Achilles tendon (Sigma-Aldrich, St. Louis, MO, USA), Viscolma collagen suspension dissolved at 12% (Viscofan BioEngineering, Weinheim, Germany), nanohydroxyapatite (nanoXIM) aggregates (Fluidinova S.A., Maia, Portugal), 1-ethyl-3-(3-dimethyl aminopropyl) carbodiimide hydrochloride (EDC) and N-hydroxysuccinimide (NHS) (Fluka, Buchs, Switzerland), hydrochloric acid (HCl) (Merck KGaA, Darmstadt, Germany), Vancomycin (50 mg/mL, HIKMA Farmacêutica, S.A., Sintra, Portugal) and Gentamicin (40 mg/mL, Labesfal—Laboratórios Almiro, S.A., Portugal), were kindly provided by Artur Salgado S.A. (Maia, Portugal). 

Alamar blue dye (resazurin), magnesium chloride (MgCl2), 4′-6-diamidine-2-phenylindole (DAPI), formaldehyde 4% and Triton X100 were purchased from Sigma-Aldrich (Sigma-Aldrich, St. Louis, MO, USA); the dimethyl sulfoxide (DMSO) was obtained from Merck (Merck KGaA, Darmstadt, Germany). Dulbecco’s modified eagle medium (DMEM), fetal bovine serum (FBS), fungizone, penicillin-streptomycin and trypsin were purchased from Gibco (Thermo Fisher Scientific, Waltham, MA, USA). The DC™ protein assay was purchased from Bio-Rad. The Alexa fluorconjugated phalloidin 594 and the Quant-iT™ Picogreen^®^ DNA assay kit were purchased from Invitrogen (Thermo Fisher Scientific, Waltham, MA, USA).

### 2.2. Preparation of the Collagen-NanoHA Scaffolds 

For batch A, Achilles tendon bovine collagen type I was dissolved using a diluted solution of HCl (10 mM) and kept at 4 °C. For batch B, the 12% Viscolma collagen suspension was used. In order to remove all of the lumps from the solutions, they were homogenized at 20,000 rpm (Ultra Turrax T25, IKA) at 4 °C for about 1 h and 30 min. In order to produce the samples, the collagen solution was mixed using a peristaltic pump, first with a HCl and nanohydroxyapatite solution, and after homogenization with a HCl, EDC (40 mM) and NHS (20 mM) solution. The collagen and nanoHA were in 50:50 % *w/w* proportions. The resultant solution filled two sizes of glass molds (5 and 10 cm of diameter) and was kept in the freezer for 24 h. Finally, the samples were dried using a freeze-drier (Labconco) for 24 h (−80 °C). Some samples were submitted to sterilization through e-beam irradiation (15 kGy). Eight different materials were studied, before (A and B) and after (A’ and B’) their e-beam irradiation, with 5 and 10 cm diameters. 

### 2.3. Characterization of the Cryogels

#### 2.3.1. Scanning Electron Microscope Analysis

A scanning electron microscope (SEM, FEI Quanta 400FEG, Hillsboro, ON, USA) was used to analyse the morphology of the samples. The samples were first attached with Araldite™ to an aluminium sample holder and then sputter-coated with palladium-gold (Bal-Tec: SCD 050) to become electrically conductive and be analysed.

#### 2.3.2. Swelling Capacity Test

The swelling capacity test was carried out at room temperature to evaluate the ability of the scaffolds to capture and retain a solution. Samples with a cubic shape were submerged both in distilled water and aqueous phosphate-buffered saline (PBS). The study was carried out over 1 h, and the samples were weighed at the beginning of it and after each time-point. The equilibrium of the absorption—the swelling equilibrium (*Cw*)—was calculated using the formula
*Cw* = *Ws* − *Wd*/*Wd*(1)
where *Ws* corresponds to the weight of the swollen sample after the immersion, and *Wd* corresponds to the dry weight (before the immersion in water or PBS). Three samples of each type of scaffold were used, and an average was calculated and used to obtain a variation.

#### 2.3.3. Dynamical Mechanical Analysis

The dynamical mechanical analysis (DMA) assay was carried out in order to evaluate the mechanical properties of the samples under compression, and was conducted in a Tritec2000 dynamic mechanical analyser (Triton Technology Ltd., Nottinghamshire, UK). Samples of parallelepiped form with 5 mm thickness and around 10 mm length were cut and submerged in 10 mL water for 10 min. The scaffolds were subjected to cycles of compression with frequencies varying between 0.1 Hz and 15 Hz at room temperature. The sample modulus was calculated automatically by Triton software with the sample stiffness (N/m)/geometry factor (calculated from each sample dimensions) at each frequency. An average of 5 cycles of three independent materials (same batch) were considered for the graphic plot.

#### 2.3.4. Fourier Transform Infrared Spectroscopy 

The cryogels were analysed using Fourier transform infrared (FT-IR) spectrophotometer (Perkin Elmer) in order to study their chemical composition. The analysis was carried out in the ATR mode, by compressing the samples until a clear spectrum was shown in the screen and the transmittance peaks were then registered. Every sample was analysed with a resolution of 4 cm^−1^ and an OPD of 0.2. In total, 100 scans were performed to obtain each graphic.

#### 2.3.5. In Vitro Biodegradation Analysis

The biodegradation analysis evaluated the effect of exposing different samples to simulated body fluid (prepared according to Kokubo and Takadama, 2006) [11] at 37 °C, with 200 rpm agitation for 7, 14 and 28 days. The samples were incubated in polyethylene tubes with SBF at a ratio of 1 g material per 10 mL solution, changing the solution twice a week. The biodegradation of the samples with and without sterilization was measured by weighing the sample before and after incubation in SBF.

#### 2.3.6. Vancomycin and Gentamicin Loading and Release

Vancomycin and Gentamicin were adsorbed on different Coll/nanoHA samples (A, B, A’ and B’) for 2 h at 37 °C and 120 rpm in the orbital shaker (KS 4000 IC control, IKA^®^). The samples were incubated with vancomycin and gentamicin aqueous solution. Afterwards, both solutions were removed, and every 24 h for 10 days, 200 µL of the solution was removed to determine the concentration of vancomycin or gentamicin released from the different samples, which was then replaced with 200 µL fresh PBS. The removed supernatant solution was centrifuged at 14,000 rpm for 5 min (Heraeus Fresco 21 Centrifuge, Thermo Scientific™). The vancomycin and gentamicin concentration was determined by molecular absorption spectrophotometry at 280 nm using a UV–Vis Spectrophotometer (NanoDrop^®^ ND–1000, Thermofisher Scientific, Waltham, MA, USA) and the obtained standard calibration (Vancomycin—50 to 0.005 mg/mL and gentamicin—40 to 0.004 mg/mL). All of the tests were performed in triplicate for each type of sample; 100% vancomycin release refers to the 50 mg/mL solution, and 100% gentamicin release refers to the 40 mg/mL solution.

### 2.4. In Vitro Biological Studies

#### 2.4.1. Cell Culture

The L929 (ATCC), MG63 (ATCC) and HBMSC (human bone marrow stromal cells—Hospital São João, Portugal) cells were maintained in Dulbecco’s modified eagle medium (DMEM, Gibco) supplemented with 10% (*v/v*) fetal bovine serum (FBS) (Gibco), 1% (*v/v*) fungizone and 1% (*v/v*) penicillin-streptomycin (Gibco). The cells were kept in a cell culture incubator (Binder, Tuttlingen, Germany) at 37 °C and 5% carbon dioxide (CO_2_), in a humidified atmosphere. After a cell confluence of 90% on a 75 cm^2^ T-flask (Nunc), which occurred every 3–4 days, the cells were detached using trypsin (0.5%, Gibco) as the dissociation reagent to be seeded on the cell culture plates or within the scaffolds. 

#### 2.4.2. Cellular Metabolic Activity

The resazurin (Alamar blue dye) assay was performed to study the direct cytotoxicity of the samples when they were in contact with MG63 and L929 cells, through its metabolic activity. The samples were sterilized as described before and incubated with complete DMEM for 30 min at room temperature in 24-well plates (no tissue culture). The medium was removed and 20 μL of the cells (3 × 10^5^ cells) was added to each sample. The plates were incubated for 2 h at 37 °C and the samples were then covered with complete cell culture medium. The samples were incubated for 1, 7, 14 and 21 days, and the medium was changed every three days. After the removal of the medium, 1 mL of a solution of 10% resazurin (0.1 mg/mL, Sigma-Aldrich) was added and protected from the light, and the plates were incubated for 3 h at 37 °C. A solution of 100 μL of each well was transferred to a black 96-well plate (triplicates) and the fluorescent intensity was measured in a fluorometer (Synergy Mx, BioTek) at 530 nm for excitation and at 590 nm for emission. After that, the samples were washed twice with PBS and re-incubated with fresh complete cell culture medium. The seeding of L929 and MG63 cells in Tissue Culture Polystyrene (TCPS) with supplemented DMEM was also performed to analyse the cells’ metabolic activity in the absence of the collagen-nanoHA scaffolds.

#### 2.4.3. Alkaline Phosphatase (ALP) Activity

After the resazurin assay, samples with MG63 and HBMSC cells from day 14 were washed twice with PBS and incubated with milli-Q water for 1 h at 37 °C, and then freeze at −80 °C for 1 h. After that, the samples were cut into small pieces, homogenized in the vortex for 1 min and centrifuged (Centrifuge 2-16PK, Sigma) at 2000 rpm for 5 min. The supernatant (20 μL) was then transferred to a 96-well flat-bottom plate (triplicates), and 200 μL ALP substrate was added according to protocol published before [12]. The absorbance was measured in a microplate reader (Synergy Mx, BioTeK) at the wavelength of 405 nm, and the p-nitrophenol was quantified. In order to correlate the amount of ALP to the total quantity of protein present in the samples, the total protein content was measured using Lowry’s method, according to the manufacture’s recommendations (DC™ protein assay, Bio-Rad). Finally, the amount of ALP was calculated using the equations of both the calibration curves referred to before, and it was expressed in nmol per minute per mg of protein. 

#### 2.4.4. DNA Quantification Assay

Samples from the HBMSC culture after 7, 14 and 241 days were processed as described in the previous assay, and the supernatant was used to quantify the amount of DNA present in the samples, according to the manufacture’s recommendations (Quant-iT™ Picogreen^®^ DNA assay, Invitrogen, UK). Briefly, a suspension of each sample (10 μL) was pipetted into a black 96-well flat-bottom plate (triplicates), as well as a blank composed of milli-Q water, and 90 μL TE buffer (1×) was added in triplicate. High-range standard solutions (1 ng/mL to 1 μg/mL) were prepared from a cDNA stock solution (2 μg/mL). Each standard solution (100 μL) was pipetted in triplicates to the same plate of the samples. Afterwards, 100 μL PicoGreen reagent was added to every well (samples and standards) and the plate was incubated for 5 min whilst protecting it from the light. The fluorescent intensity was measured in a fluorimeter (Synergy Mx, BioTek) with an excitation wavelength of 480 nm and an emission of 520 nm.

#### 2.4.5. Confocal Laser Scanning Microscopy

Confocal laser scanning microscopy (CLSM, Leica SP2 AOBS SE camera) was used to study the morphology of the MG63 cells after 14 days of being incubated with the samples before and after e-beam irradiation. The samples were incubated in paraformaldehyde 4% (Sigma-Aldrich) for 30 min at room temperature. After that, the materials were incubated with Triton X100 solution (0.1%, Sigma-Aldrich) for 30 min at room temperature and washed twice with PBS. The samples were then incubated for 30 min. The cells’ cytoplasm (phalloidin) was stained with Alexa fluor conjugated phalloidin 594 (1:400, Invitrogen) and the nuclei were stained with DAPI (4′-6-diamidine-2-phenylindole, 1 µg/mL, Sigma-Aldrich) for 5 min. The samples were washed twice in PBS evaluated by confocal laser microscopy. The images were captured using excitation lasers of 405 nm and 594 nm.

### 2.5. In Vivo Biological Studies

#### 2.5.1. Animal Model Protocol

E-beam-sterilized collagen/nanohydroxyapatite (#A’) prototype was implanted in the tibia of 14 male, 13-week-old rabbits (UTAD, Portugal). The study was carried out in accordance with the Animal Studies Ethics Committee and fulfilled all of the legal requirements (approved by the Animal Welfare and Ethics Body, UTAD, Portugal and Direção Geral de Alimentação e Veterinaria (DGAV) (approval 010532/2018). The surgical procedures were performed under standard aseptic conditions. One bone defect was created with a 6 mm diameter for a study implant (Coll/ nanoHA membrane, #A’). The rabbits were sacrificed 5 and 15 weeks after the prototype implantation, and the proximal tibia was collected for analysis.

#### 2.5.2. Histological Analysis

All of the samples were removed and fixed in 10% neutralized buffered formalin for 3 days, decalcified after microCT analysis with 10% formic acid, and then processed for histology. The processed samples were embedded in paraffin and sectioned longitudinally. The slides were stained with hematoxylin-eosin (H&E).

### 2.6. Statistical Analysis 

The data from the in vitro assays with cell cultures was presented as the mean ± standard deviation, and was analysed using a two-way ANOVA test. The differences between the samples were considered statistically significant when *p* < 0.05. 

## 3. Results

### 3.1. Characterization of the Cryogels

#### 3.1.1. Scanning Electron Microscope Analysis

A scanning electron microscope (SEM) was used to analyse the morphology of the scaffolds. In Figure 1, the microscopy images show that all of the scaffolds presented interconnected pores of different sizes. The materials from batch B, with Ø 10 cm, both irradiated and non-irradiated, presented a higher number of pores, and because of that it was hard to handle, being damaged by manipulation. It was also possible to observe that irradiation did not have an impact on the scaffolds’ porosity, as no major differences were observed during the analysis. Between the largest (Ø 10 cm) and smallest (Ø 5 cm) size membranes, there were no significant differences in the pores’ dimensions. In the images (Figure 1), spherical aggregates of nanoHA could be observed in the scaffolds, probably due to problems in the homogenization step of the ceramic powder during the production of the cryogels.

#### 3.1.2. Swelling Capacity Test

The swelling capacity test was performed in order to evaluate the ability of the scaffolds to capture and retain solutions (distilled water and PBS) for 60 min. In Table 1, we can observe the behaviour of the collagen/nanoHA scaffolds A and B (with 5 and 10 cm, before and after e-beam irradiation) immersed in distilled water and PBS after 2, 15 and 60 min.

For all of the samples, the uptake of water showed an increase in the first 2 min. The *Cw* value did not show major differences until it reached 15 min, the time-point previously indicated in the bibliography as being the equilibrium [12]. The values were similar until the last time-point (60 min). Because the values of *Cw* were maintained from minute 2 until 60 min, it is possible to conclude that the equilibrium was reached at 2 min. It is possible to see that the samples that showed the lower values of the swelling coefficient were the ones from batch B, especially the samples with Ø 10 cm, which had the lowest value when compared to all of the tested samples. In general, the largersamples (Ø 10 cm) had a higher water uptake when compared to the smaller materials (Ø 5 cm). Regarding the PBS uptake, the results of the absorption coefficient also showed a difference between the scaffolds from different batches, with batch B with Ø 5 cm having the lowest value of *Cw*. The values for the *Cw* were very similar to the same materials evaluated in the presence of water, and the larger samples were also the materials with higher *Cw* values. Comparing the results of the samples before (A and B) and after e-beam irradiation (A’ and B’), it is not possible to observe significant change of the water uptake after irradiation. On the contrary, the PBS uptake by the irradiated scaffolds was lower (B and B’), with the non-irradiated samples having the higher values of *Cw*.

#### 3.1.3. Dynamical Mechanical Analysis

The dynamical mechanical analysis (DMA) assay was performed in order to evaluate the mechanical properties of the scaffolds. All of the samples showed a rise of E’ as the frequency increased, until 10 Hz. It is also possible to observe that the scaffolds from batch A with Ø 10 cm had higher modulus values when compared with all the other evaluated samples (Figure 2A.B). With respect to tan delta results, all tested samples showed similar results before and after sterilization, except for scaffolds A with 5 cm that showed a significant increase at 15 Hz. Comparing the results from the materials before and after irradiation, it is possible to observe that samples A and B reached lower values of E’ and lower values of Tan delta in higher frequencies.

#### 3.1.4. Fourier Transform Infrared Spectroscopy 

Fourier transform infrared spectroscopy (FT-IR) ATR determine qualitative and quantitative features of IR-active molecules in inorganic and organic samples that are showed in a spectrum. FT-IR was used to characterize the chemical groups present in the cryogels in order to observe their composition (Figure 3). The characteristic bands of collagen can be seen at 1648 cm^−1^, which refers to the amide I (C=O stretching); 1550 cm^−1^, corresponding to the amide II (N-H deformation); and 1239 cm^−1^, which is the amide III (N-H deformation). The peaks of the nanoHA, which result from the vibration of the phosphate groups (PO_4_^−3^), are also represented at 1031 cm^−1^ (V3 and V1 mode vibrations of PO_4_^−3^), 962 cm^−1^ (V3 and V1 mode), 602 cm^−1^ (V4 mode) and 564 cm^−1^ (V4 mode). The band of carbonate (CO^−3^, V2 vibration) can be observed at 875 cm^−1^. 

#### 3.1.5. In Vitro Biodegradation Analysis 

The mass loss of the different collagen/nanoHA samples with and without irradiation was determined after soaking in PBS (Figure 4). According to the results, sample A’ had a slightly higher mass loss after 7 and 14 days, although no significant differences were detected between any of the irradiated samples for time of observation.

#### 3.1.6. Vancomycin and Gentamicin Loading and Release

The vancomycin and gentamicin release profile of samples A, B, A’ and B’ is shown in Figure 5. For all of the materials, the obtained vancomycin release profile had a high initial burst and a sustained release (3 days) followed by 2–3 days of decrease at concentrations which were always above the minimum inhibitory concentration (MIC) for MRSA (MIC = 2 μg/mL) until 10 days [13]. For sample A’, there is significant decrease in the amount of antibiotics released after 2 days when compared to the other samples. However, at day 6, the released antibiotic concentrations were the same for all of the tested materials. Similar behaviour were observed on the gentamicin release profiles for all of the samples, but at day 8, the antibiotic concentration did not reach the limit of detection (concentration values below ≤0.5 μg/mL).

### 3.2. In Vitro Biological Studies 

#### 3.2.1. MG63 and L929: Resazurin Assay

The Alamar blue assay is used to assess cell viability. In this work, the resazurin assay was performed in order to study the direct cytotoxicity of the samples A, B, A’ and B’ in direct cultured with mice fibroblasts (L929) and human osteoblasts-like cells (MG63—Figure 6).

The fluorescence intensity of MG63 cultured in direct contact with different scaffolds (A, B, A’ and B’) over 2 weeks is presented in Figure 6A,B. During the first week, the cellular viability of the MG63 cells was lower when compared to the day 1 results for scaffolds B and B’. On day 14, the cell viability increased for all of the tested samples. Scaffolds B (Ø 10cm and Ø 5cm) and B’ (Ø 10cm and Ø 5cm) showed higher cellular viability when compared to samples A and A’ after 14 days (Figure 6A,B). Figure 6C,D represents the fluorescence intensity of L929 that was cultured in the different materials (A, B, A’ and B’) for 1, 7 and 14 days. A higher fluorescence intensity is related to a higher metabolic activity and thus a higher cell viability. It is possible to observe that the cell viability increased until day 7, when it reached the highest values. After 14 days, L929′s viability was very similar in all of the evaluated materials. Therefore, the results obtained with MG63 in contact with irradiated materials (A’) showed a lower metabolic activity after 14 days of culture when compared with non-irradiated samples (A).

#### 3.2.2. MG63: Alkaline Phosphatase (ALP) Activity

The assessment of the alkaline phosphatase activity was evaluated with osteoblast-like cells (MG63) after 14 days of culture within materials that were and were not e-beam irradiated (Figure 7). ALP is a parameter of early cell differentiation, and is therefore an early indicator of the expression of the osteoblastic phenotype. As shown in Figure 7, the ALP activity was higher on scaffold B’ when compared to the other samples, both with Ø 5 and 10 cm, although it was not statistically significant. 

#### 3.2.3. Confocal Laser Scanning Microscopy

Confocal laser scanning microscopy (CLSM) was used to study the morphology of the MG63 cells after 14 days of culture within samples that had and had not been e-beam irradiated. The CLSM images show that the cells had adhered to the scaffolds’ surface (proximally 100 µm deep), but it was not possible to observe the cells in the inner part of the scaffolds (over 150 µm), and in some samples the cells were detected only at the edges of the material. It was also possible to distinguish cells in two different morphologies: one round and agglomerated, and another spread and well adhered. Comparing cryogels A,B, A’ and B’, it was possible to observe a higher cell number for most of the samples in materials after sterilization (Figure 8). 

#### 3.2.4. HBMSC: Alkaline Phosphatase (ALP) Activity and DNA Quantification Assay

The results in Figure 9A showed a higher HBMSC DNA concentration from day 7 to day 14, remaining similar concentration until 21 days. The increase in the DNA concentration was higher for samples B and B’after 14 days. These findings corroborated the higher cellular metabolic activity results in the same materials with MG63 cells (Figure 6), in which the cells showed a higher cellular metabolic activity until 14 days. The HBMSC cells’ ALP activity results show lower enzymatic activity when compared to selected values in studies performed with MG63 cells, but an increase in ALP activity was also observed with the time of culture. In Figure 9B, it is possible to observe a significant increase in the ALP enzyme activity only for scaffolds A, B and B’. In order to assess the scaffolds’ osteoinductive capacity, HBMSC were cultured in the different materials in the absence of dexamethasone (osteogenic inducer). The expression of the genes from HBMSC cells was quantified in real time by qPCR, which showed higher levels of the bone morphogenic protein type 2 (BMP-2) production marker, with the B’ sample being significantly higher among the different samples tested. However, low levels of osteocalcin (OC) (Figure 9D) were quantified in all of the samples evaluated. The study demonstrates that the cells are in an earlier differentiation state for the osteogenic genotype (pre-osteoblasts), producing higher amount of proteins (Figure 9C), but they are not yet able to mineralize the extracellular matrix (the gene expression for osteocalcin—Figure 9D).

### 3.3. In Vivo Biological Studies 

The researchers involved in the animal handling were FELASA accredited and DGAV certified for animal experimentation. The bone defects were totally filled with the implanted scaffolds. Histological analyses of the scaffolds inside the bone tissue, using hematoxylin-eosin staining, indicated a uniform distribution of the connective tissue throughout the A’ scaffold after 5 weeks (Figure 10B,C) without acute inflammatory response, and the presence of small collagen/nanoHA fragments from the scaffold structure. The material was integrated within the surrounding tissue, and some disorganized bone was formed on the board of the implant after 5 weeks (Figure 10A–C).

Similar bone tissue growth was observed after 15 weeks with the biocomposite scaffold, probably due to the fact that bone cells had reached the innermost parts of the materials. The composite scaffolds showed higher bone tissue ingrowth and vascularization after 15 weeks of implantation (Figure 10D,F).

## 4. Discussion

The main tasks of the work included the physical, chemical, mechanical and biological characterization of the scaffolds produced on a large scale as industry prototypes. In the SEM analysis (Figure 1) showed ceramic nanoparticles aggregates within the scaffolds, probably due to problems in the homogenization step of the ceramic powder during the production of the cryogels. These aggregates are similar to the ones observed in the nanoHA granules studied by Laranjeira et al. [14]. The pores observed in all of the samples were very heterogeneous, with sizes varying from 50 µm to 900 µm, as measured by imaging software (Image J), which gave the materials’ both microporosity and macroporosity. This was an important characteristic, as pores with sizes larger than 100 µm should be essential for cell seeding and tissue ingrowth, whereas pores with diameters higher than 140 µm are important to promote angiogenesis [15,16,17,18,19,20]. From the results of both tables, it is possible to observe that the non-irradiated scaffolds presented larger pore sizes. E-beam irradiation have an important impact in synthetic polymer degradation, inducing chain scission, but in this work, it does not have the same effect on the natural polymer crosslinking, so it should not have an effect on the pore sizes [21]. Regarding the comparison between the scaffolds with Ø 10 cm and Ø 5 cm, no correlation could be established, as the irradiated samples showed larger pores and, in contrast, the non-irradiated samples showed smaller pores. In previous studies, scaffolds of collagen/nanoHA 50:50% *w/w* produced by Rodrigues and collaborators also showed a heteroporous morphology, with pores of an average size of 74.39 ± 49.05 µm, a maximum size of 322.09 µm and a minimum of 12.16 µm [12]. Collagen-nanoHA (50:50% *w/w*) scaffolds produced by Sionkowska et al. also showed a heteroporous morphology, with pore sizes varying from 50 to 150 µm [22].

Because the values of the *Cw* were maintained from minute 2 until minute 60, it is possible to conclude that the equilibrium was reached at 2 min, as a similar result was observed in Jain et al. studies [23]. It is possible to see that the samples that showed lower values of the swelling coefficient were the ones from batch B, especially the samples with Ø 5 cm. The fastest water uptake, shown after the first 2 min of our study, is closely related to a higher interconnectivity of the pores and the hydrophilicity of the material, which is the objective of the scaffold for clinical applications, as it allows a higher diffusion of molecules, nutrients, gases and fluids throughout the material [23]. In comparison, the hydrogels’ water uptake is a slower process, as it is dependent on the water diffusion [23]. Furthermore, scaffolds that retain a higher amount of water are associated with the presence of larger pores [24,25]. This higher retention of water and PBS will result in the swelling capacity of the scaffolds, which will expand the material structure that is available for cell migration and adhesion, because of a closer contact with the surrounding tissue. However, the higher retention of the solvent can also decrease the mechanical properties of the scaffold by decreasing its elastic strength [26]. Therefore, the objective for the final clinical application is a structure with interconnected pores that will allow an equilibrium between the swelling capacity and the mechanical properties, without compromising any of them. In this study, these properties correspond to the materials with lower *Cw* values. The samples from batch B showed better swelling capacities with both water and PBS, before and after irradiation, closer to the results observed in previous studies, which were 18.54 (water) and 17.56 (PBS) after 15 min for collagen-nanoHA (30:70% *w/w*) scaffolds. As for the collagen-nanoHA (50:50% *w/w*) scaffolds, the values were higher, at 28.63 (water) and 30.04 (PBS) [12]. The collagen used for batch B was from a different source in comparison to the other cryogel (A). In this batch, the purchased collagen was already dissolved, so the collagen stability (degradation) could have been compromised because the solution pH was very acidic (pH < 2). The differences between batches A and B could also be related to the difficulties in performing an adequate homogenization of the collagen with nanoHA in the reactor during the materials’ production. Thus, samples with a higher concentration of nanoHA are connected to a lower swelling ratio. This could be explained by the lower hydrophilicity of hydroxyapatite, and by the fact that, when combined with collagen, the calcium and phosphate of hydroxyapatite will bind to the hydrophilic groups of collagen (COOH and NH_2_), resulting in a decrease in the overall hydrophilicity of the sample [22]. Although the results were similar to previous studies [12], the protocol could be revised and optimized in order to obtain scaffolds with lower values of *Cw* and higher mechanical properties. 

In the scaffolds A, B, A’ and B’, as in the materials produced by Rodrigues and co-workers, the peak of amide I appears at 1648 cm^−1^ and not at 1658 cm^−1^ when the collagen is not combined, probably due to the interaction of nanoHA and collagen through carbonyl groups [27,28].

The scaffolds before and after the sterilization process were evaluated under a dynamic compressive stress state. The DMA analysis showed the variation of the storage modulus (E’) for the samples, which refers to the elastic component of a material [29], with respect to the frequency. The higher the storage modulus, the higher the stiffness of the material. By comparing the results from the materials before and after irradiation, it is possible to observe that samples A and B reached higher values of E’ and lower values of Tan delta in higher frequencies. These discrepancies between all of the material batches may be a result of a heterogeneity of the nanoHA solution dispersion into the collagen structure. Higher concentrations of nanohydroxyapatite will lead to a decrease in the loss factor and an enhancement of the storage modulus [12,30], showing the higher stiffness of the material. Nevertheless, in the same batch, the results from both of the evaluated parameters (E’ and Tan delta) were also different before and after the e-beam irradiation, which could have been affected by some alteration in the collagen chains induced by energy generated by the irradiation. According to previous works, e-beam radiation induces chain polydispersity of synthetic polymer, affecting the mechanical and degradation properties, but maintaining the crosslink. This will lead to a decrease in the solubility of the polymer and its fragmentation, which can explain the differences observed in the mechanical properties of samples A, B, A’ and B’ [31]. The increase of the E’ is related to material’s higher stiffness and mechanical resistance. Rodrigues and co-workers (2013) observed a faster increase in the E’ as the frequency increases, followed by a stabilization of the modulus value. However, their study showed higher E’ results when compared with the similar materials evaluated in this work, as shown in Figure 2 [12]. Studies performed by Salgado and collaborators (2016) showed an increase of the storage modulus with the enhancement of the frequency applied to the materials. Moreover, the studies also presented by Rodrigues et al. (2013) showed an enhancement of the loss factor as the frequency increased, with values varying between 0.2 and 0.8 [12,30]. The enhancement of the tan delta shows a more viscous and less elastic behaviour of the material; as such, the ideal material’s response would be a lower value of the loss factor. Taking this into consideration, it was shown that sample A’, with Ø 10 cm, presented higher values of E’ and lower values of tan delta. Through the analysis of the results, it was also possible to observe that the larger scaffolds (Ø 10 cm) presented higher values of E’ and lower values of tan delta, and therefore the material could be used for the proposed clinical application.

The resazurin assay is a test used to infer the cell viability through its metabolic activity. Resazurin is a non-fluorescent dye that is reduced in living cells by mitochondrial enzymes to resorufin, a fluorescent dye [32]. Studies carried out by Laranjeira et al. (2010), showed a cell viability of 100% for MG63 cells cultured for 3 days on nanoHA aggregates, which were higher cell viability values compared to the results shown in this work. After 6 days of incubation, there was as increase in cell viability [14]. The higher results should be related to the material’s composition, because in these studies, unlike ours, the nanoHA was sintered and not dispersed, which is more stable and has lower degradation rate. The fibroblasts (L929) showed higher metabolic activity in all of the tested materials during the evaluated period of time, and showed similar values for both the irradiated and non-irradiated samples. However, the osteoblast-like cells (MG63) viability was mainly enhanced in the samples that undergo e-beam irradiation. This should be associated to the irradiation process, in which the energy produced by the sterilization method could induce some chemical alterations in the polymer chain’s surface, and could affect the cellular response.

The different cell behaviour (MG63) found between the different samples (A and B) should be due to a different topography of the cryogels with higher nanoHA aggregates on the surface (as observed in the SEM images—Figure 1). Rougher surfaces promote a higher proliferation rate of osteoblast-like cells when compared to smoother ones [33,34,35,36]. In the studies performed by Rodrigues and co-workers, cryogels of collagen/nanoHA 50:50 % *w*/*w* cultured with MG63 cells showed a similar cell density over the material’s surface (Confocal microscopy images–Figure 7) [12]. Therefore, considering the similarity of the published results and the figures shown in this work, we can observe a high influence of the material characteristics on MG63 cell behaviour, favoring the cellular proliferation rate. The ALP activity results for the MG63 culture (Figure 5) showed higher enzyme activity when compared to the values in studies conducted with similar biomaterials [12,14]. This higher activity could be induced by a higher nanoHA content of the B’ samples, because this ceramic had been demonstrated to have an influence on the enhancement of ALP activity of MG63 cells [37,38]. The ALP had a lower activity in sample A, probably due to the fact that the cells reached a maximum confluence [39,40], which might have occurred at day 14. In the results observed in similar studies, the values for the cells’ ALP activity cultured in scaffolds of collagen/nanoHA 50:50 % *w*/*w* were 0.35 nmol/min/µg for a cell culture time of 14 days [12]. Previous studies by Laranjeira et al. (2010) showed an increase in the ALP activity from day 3 to day 6 for the cells seeded in nanoHA granules [14]. 

The HBMSC proliferation rate was estimated by the quantification of DNA, and it was possible to observe that all of the evaluated scaffolds promoted a high proliferation rate until day 21. In addition, the total DNA content at 14 days of culture in samples A’ and B’ was significantly higher when compared to day 7, and remained with a slightly higher concentration after 21 days. Therefore, in these in vitro tests with HBMSC, scaffolds A, B and B’ exhibited increasing levels of differentiation by higher ALP activity and bone morphogenetic protein 2 (BMP-2) gene expression levels. Indeed, we observed an increasing of cellular proliferation (Figure 9A) over the studied period; this behaviour was a positive result that should end up to a normal tissue growth for the proposed clinical application.

After 5 weeks, Coll/nanoHA scaffolds was removed from the rabbit tibia, the histological evaluation was performed and showed that the materials’ structure was partially present. The maintenance of the composite structure should be related to the chemical bonding of the collagen fibrils allowed by the crosslinking step (EDC/NHS). The histological findings indicate that Coll/nanoHA supported tissue ingrowth and new micro-vascularization. Importantly, in the context of bone ingrowth, the presence of connective tissue that replaces the hematoma during the early stages of tissue repair will favor the intramembranous ossification enhancing the bone repair. Previous studies showed that the biocomposite scaffold cell-binding domains could promote the in vivo migration and proliferation [12,30] of animal fibroblasts, an important cell population that will promote host cell migration and proliferation. Increased tissue growth was observed for the bone implants, similarly to *the* in vitro analysis of the MG63 and HBMSC cells (Figure 6A,B and Figure 9A). The total bone ingrowth area was calculated by the total volume fraction (microCT) after 5 and 15 weeks (Figure 10G). The results show that the A’ scaffolds induced the continuous growth of connective tissue within the porous structure after 5 weeks, along with new bone growth at the defect border. However, after 15 weeks, the defect area was almost filled with disorganized new-bone tissue, replacing the total scaffold area (material’s biodegradation) (Figure 10E,F).

## 5. Conclusions

This work had as its main objective the evaluation of a scaled-up productive process for the development of a biomaterial based on collagen and nanohydroxyapatite at the industrial prototype level. The work included the evaluation of the material’s safety and reproducibility, in order for it to be used as a medical device in the orthopaedic area. The materials showed an ability to absorb and retain solvents, as well as adequate mechanical resistance and viscoelastic properties, and the collagen and nanohydroxyapatite presence in the scaffolds was confirmed. The cryogels had a heteroporous morphology with microporosity and macroporosity, which are essential conditions for the interaction between the cells and materials, and consequently to promote bone regeneration. For the biological studies, metabolic activity assays were performed and showed no evidence of the cytotoxicity of the cryogels when in they were in contact with fibroblasts (L929), osteoblast-like cells (MG63) and human mesenchymal stromal cells (HBMSC). In fact, the scaffolds promoted the proliferation of the cells, as well as the expression of the osteoblastic phenotype (MG63 cell line and HBMSC). Confocal laser scanning microscopy (CLSM) corroborate the cytotoxicity results, showing no differences in the cell behavior to the different batches using collagen from Sigma-Aldrich and Viscolma. In this work, no significant differences were observed between the larger (Ø 10 cm) and smaller (Ø 5 cm) scaffold dimensions, or between the e-beam irradiated and non-irradiated materials. Based on the bone tissue’s in vivo response, it is possible to conclude that the developed biocomposite scaffolds based on collagen and HA nanoparticles did not show a chronic inflammatory response; instead, the biocomposite induced intense bone tissue formation after 15 weeks, and could have a significant therapeutic impact for bone regeneration in the coming future.

## Figures and Tables

**Figure 1 materials-14-04130-f001:**
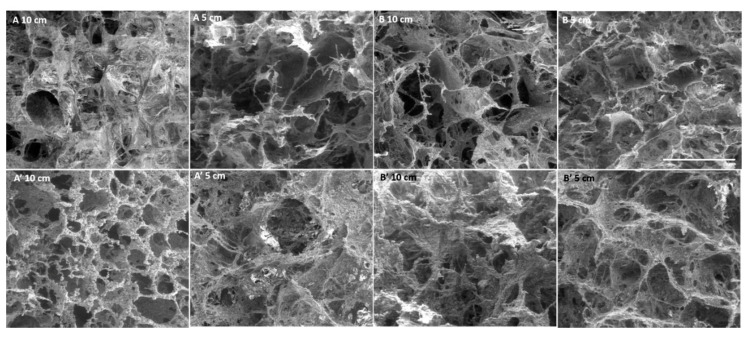
SEM images of Coll/nanoHA scaffolds A–B and A’–B’. Scale bar: 500 µm.

**Figure 2 materials-14-04130-f002:**
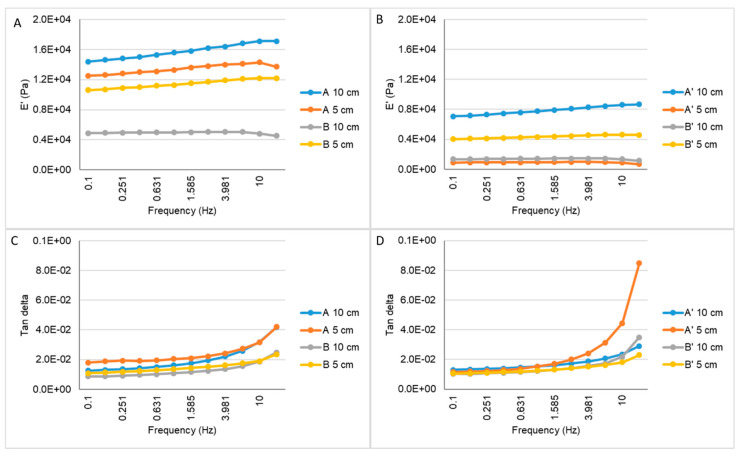
Storage modulus (**A** and **B**) and tan delta (**C** and **D**) under dynamic compression solicitation versus the increasing frequency, ranging from 0.1 to 10 Hz for Coll/nanoHA scaffolds produced with different collagens before (A and B) and after sterilization (A’ and B’).

**Figure 3 materials-14-04130-f003:**
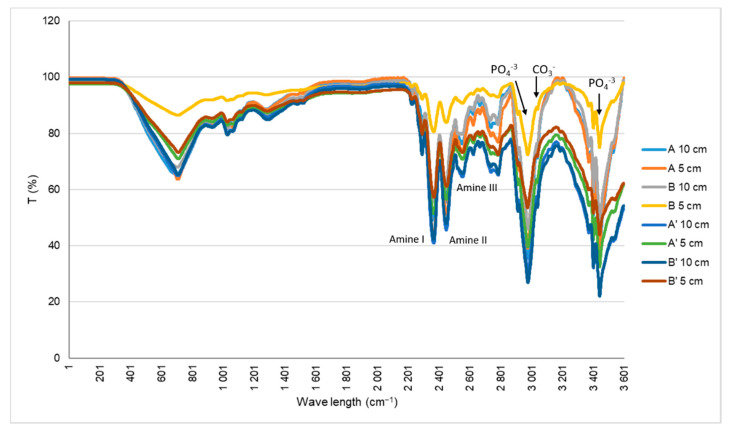
FT-IR spectra of the Coll/nanoHA scaffolds produced with different collagens before (A and B) and after sterilization (A’ and B’).

**Figure 4 materials-14-04130-f004:**
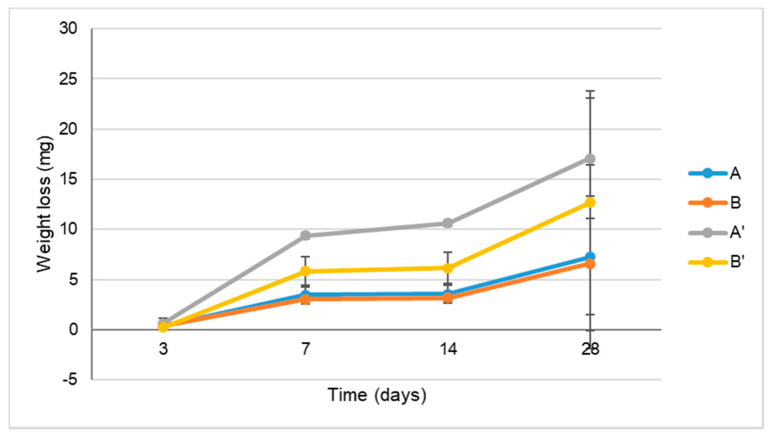
Mass loss after degradation in SBF of the collagen/nanoHA samples before and after sterilization, for different periods.

**Figure 5 materials-14-04130-f005:**
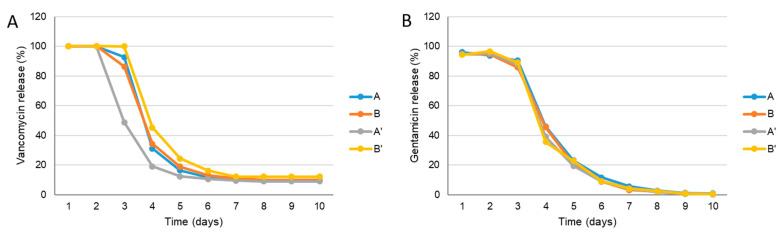
Vancomycin (**A**) and gentamicin (**B**) concentrations present in the solution released from Coll/nanoHA samples with and without sterilization, at different time points until 10 days.

**Figure 6 materials-14-04130-f006:**
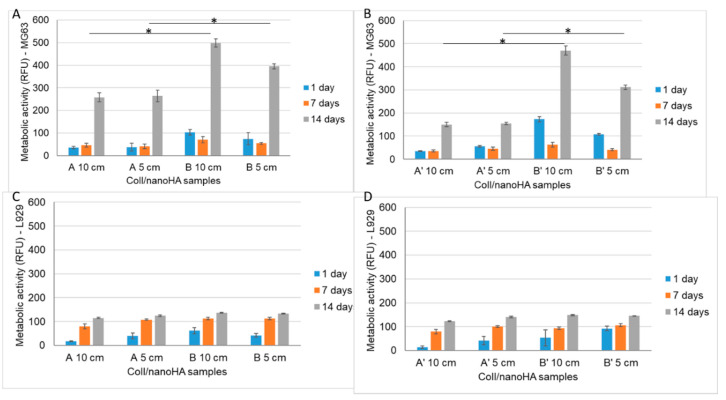
Metabolic activity of MG63 (**A**,**B**) and L929 (**C**,**D**) when in direct contact with samples A, B, A’ and B’ after 1, 7 and 14 days of incubation, as estimated by a Resazurin assay. The control was carried out in TCPS, and was considered to have a cell viability of 100%. Statistical differences between samples from different batch, * *p* < 0.05.

**Figure 7 materials-14-04130-f007:**
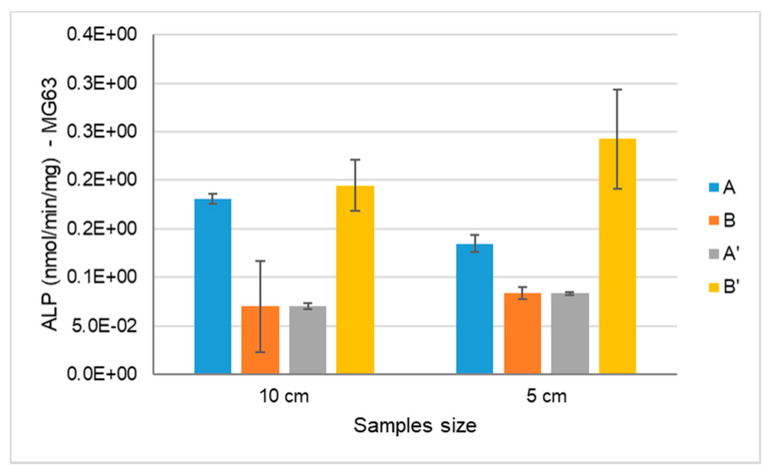
ALP activity of human osteoblast-like cells (MG63) cultured within samples A, B, A’ and B’ for 14 days.

**Figure 8 materials-14-04130-f008:**
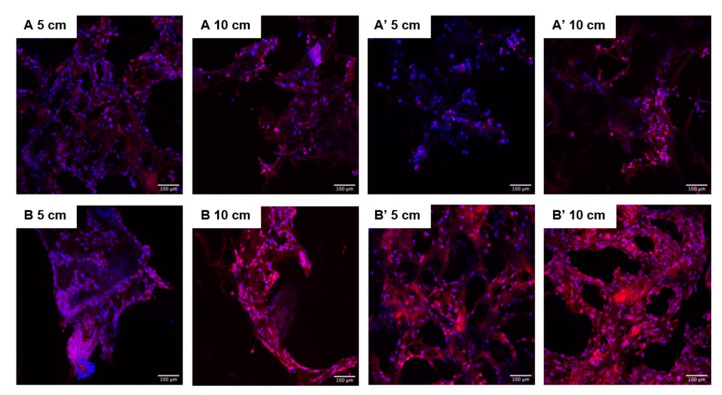
CLSM images of MG63 cultured for 21 days on scaffolds A, B, A’ and B’ with different sizes. Scale bar: 100 µm.

**Figure 9 materials-14-04130-f009:**
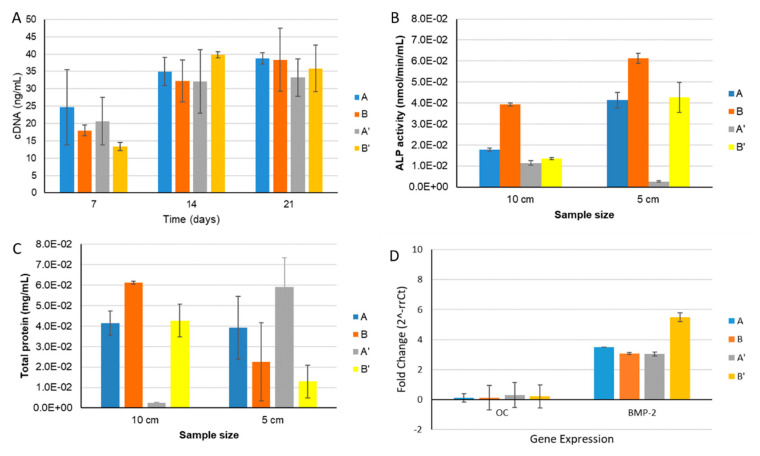
(**A**) Evaluation of the cell proliferation by the DNA extraction method from cell cultures in different samples, irradiated or not (A, B, A’ and B’), after 7, 14 and 21 days. (**B**) ALP activity and total protein content (**C**) of human bone marrow stromal cells (HBMSC) cultured within samples A, B, A’ and B’ for 14 days. (**D**) qPCR of the HBMSC osteogenic gene expression (osteocalcin–OC; Bone morphogenetic protein 2–BMP-2) after culturing on samples A, B, A’ and B’ after 14 days. The 2^-rrCt method was used with the expression of the GAPDH gene as an endogenous reference. cDNA from HBMSC cells grown in TCPS (P4) were evaluated as a negative control (Ref-1).

**Figure 10 materials-14-04130-f010:**
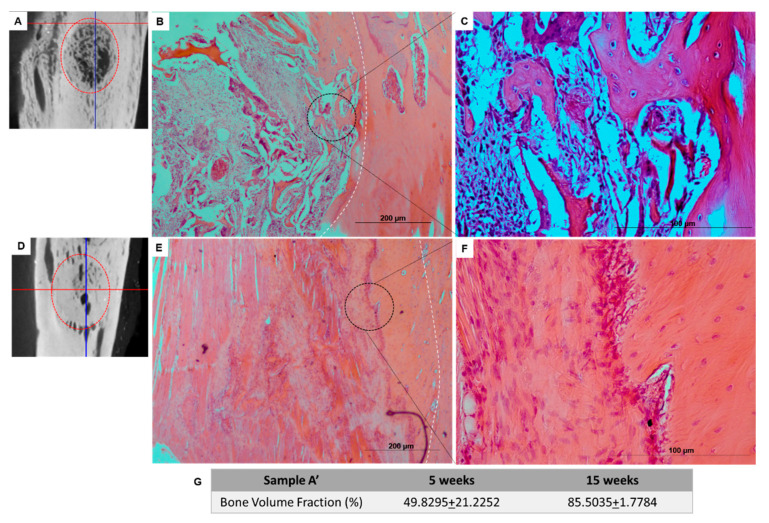
(**A**,**D**) MicroCT scan 3D reconstruction images of Coll/nanoHA implants (A’) in the bone tissue of rabbits’ tibia after 5 (**B**,**C**) and 15 weeks (**E**,**F**). (**G**) Table with the percentage of the volume fraction calculated by microCT analysis after 5 and 15 weeks.

**Table 1 materials-14-04130-t001:** Swelling kinetics of the collagen/nanoHA scaffolds before and after sterilization (A, B, A’, and B’) with different sizes in PBS buffer and distilled water.

Samples	Swelling Ratio (Cw)					
	H_2_O			PBS		
	2 min	15 min	60 min	2 min	15 min	60 min
A (5 cm)	25.37 ± 1.60	23.53 ± 2.15	23.74 ± 1.58	24.73 ± 0.77	24.93 ± 1.40	25.51 ± 0.71
A (10 cm)	27.01 ± 0.12	26.85 ± 1.10	27.73 ± 1.27	28.69 ± 0.99	27.82 ± 1.67	28.14 ± 1.94
B (5 cm)	15.07 ± 0.15	15.03 ± 0.34	15.26 ± 0.32	15.17 ± 0.40	16.28 ± 0.57	16.44 ± 1.11
B (10 cm)	17.51 ± 0.74	16.48 ± 0.58	16.68 ± 0.52	17.30 ± 0.87	16.74 ± 1.49	14.38 ± 1.66
A’ (5 cm)	23.29 ± 1.48	23.78 ± 0.83	24.40 ± 0.55	24.24 ± 1.92	27.00 ± 2.12	24.13 ± 1.87
A’ (10 cm)	24.24 ± 1.17	23.91 ± 1.21	25.34 ± 0.78	25.67 ± 0.52	26.33 ± 1.28	24.79 ± 0.72
B’ (5 cm)	15.54 ± 2.55	14.21 ± 4.85	14.32 ± 5.07	14.67 ± 0.12	13.82 ± 0.08	11.97 ± 1.79
B’ (10 cm)	13.90 ± 0.93	14.84 ± 0.29	14.68 ± 1.39	16.83 ± 2.16	13.88 ± 1.51	15.57 ± 1.13

## Data Availability

The data presented in this study are available within the article.

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
