# Peer review of "Translational Research for Orthopedic Bone Graft Development"

_materials, 2021, doi:10.3390/ma14154130_

Round 1

Reviewer 1 Report

Comments to the authors

Aiming at bone regeneration applications, scaffolds based on natural collagen and synthetic nanohydroxyapatite were developed and showed adequate mechanical and biological properties in the manuscript. The materials production process upscale was optimized, and the obtained prototype showed reproducible properties from the bench development and could be able to be commercialized.

However the following issues should be addressed before it could be published.

Detailed Comments:

  1. This text should be checked carefully. There are several format wrongs in the text, for example “4oC” in line 88, “4 cm-1” in line 121, “bafter2 days” in line 309 and “Figure 8: CLSM images of MG63 cultured for 21 days on scaffolds A and B and A’ and B’. Scale bar: 100 μm” in line 371.
  2. The format of title 2.6 “Statistical analysis” should be same as others secondary heading.
  3. There is no anything in line 227, and can be deleted.
  4. The result of experiment of “Swelling capacity test” and “Dynamical Mechanical Analysis” are got conclusion at room temperature, should do comparative experiment at temperature 37℃.
  5. The content of line 257 have no significance, it should be deleted or replaced.
  6. The format of digital of table 1 is not tidy which should be reorganized.
  7. “Figure 2” “Figure6” and “Figure 9” are not aligned and all of them are should be reorganized.
  8. The length of chapter “conclusion” is too long and the content can be brief.
  9. In line 325, there is writing “Fig. 5” but there should be written “Fig. 6” in fact.

Reviewer 2 Report

I'm very glad to become acquainted with such interesting research.
It was very nice to read your article, excellent! The authors made significant work, but, still, there are some minor remarks.

2.3.3 - Should add what mechanical parameters were calculated and the way (or used standard) to calculate them.
Figure 2 - MPa better use for E values. X-axis ticks better change to regular, e.g. 0.1, 0.15, 0.25, 0.4 etc.
Figure 3 - Ticks X-axis better change to regular
General note - I advise revising the centering of the figures.
